# Reaction times can reflect habits rather than computations

**Aaron L Wong[1][†]\*, Jeff Goldsmith[2], Alexander D Forrence[1], Adrian M Haith[1], John W Krakauer[1,3]**

[1]Department of Neurology, Johns Hopkins University School of Medicine, Baltimore, United States; [2]Department of Biostatistics, Mailman School of Public Health, Columbia University, New York, United States; [3]Department of Neuroscience, Johns Hopkins University School of Medicine, Baltimore, United States

**Abstract** Reaction times (RTs) are assumed to reflect the underlying computations required for making decisions and preparing actions. Recent work, however, has shown that movements can be initiated earlier than typically expressed without affecting performance; hence, the RT may be modulated by factors other than computation time. Consistent with that view, we demonstrated that RTs are influenced by prior experience: when a previously performed task required a specific RT to support task success, this biased the RTs in future tasks. This effect is similar to the use-dependent biases observed for other movement parameters such as speed or direction. Moreover, kinematic analyses revealed that these RT biases could occur without changing the underlying computations used to perform the action. Thus the RT is not solely determined by computational requirements but is an independent parameter that can be habitually set by prior experience.
DOI: https://doi.org/10.7554/eLife.28075.001

\*For correspondence:
wongaaro@einstein.edu

Present address: [†]Moss Rehabilitation Research Institute, Elkins Park, PA, United States

Competing interests: The authors declare that no competing interests exist.

## Introduction

The reaction time (RT) is arguably the most widely used measure in neuroscience and psychology for noninvasively assessing processing in the brain: it is assumed to reflect the time needed to complete the perceptual and motor-planning computations required to prepare a response (*Donders, 1969*; *Sternberg, 1969*; *Friston et al., 1996*; *Spivey, 2007*; *Sanders, 1998*). This assumption appears justified by evidence that RTs are modulated by factors such as stimulus complexity (*Kaswan and Young, 1965*; *Hick, 1952*; *Ratcliff, 2002*), stimulus-response compatibility (*Fitts and Seeger, 1953*; *Simon and Rudell, 1967*; *Simon and Wolf, 1963*), number of potential responses (*Henry and Rogers, 1960*; *Fischman, 1984*; *Christina et al., 1982*), or required response accuracy (*Fitts, 1954*; *Fitts, 1966*; *Reddi and Carpenter, 2000*). According to this assumption, any reduction in the RT should negatively affect the quality of the resulting response. Indeed, instructing participants to respond as rapidly as possible can lead to a decrease in accuracy (*Fitts, 1966*), consistent with changes in accuracy that occur through more direct manipulation of allowed preparation time (*Schouten and Bekker, 1967*; *Ghez et al., 1997*; *Stanford et al., 2010*; *Haith et al., 2016*).

However, recent evidence showed that it was possible to reduce the RT considerably before any decline in movement accuracy was observed. For example, when individuals were placed under strict time constraints, movement accuracy and task success decreased only after the RT was shortened by ~80 ms (*Haith et al., 2016*). Similarly, startle by a loud acoustic stimulus could evoke initiation of a fully prepared action ~70 ms earlier than typically observed (*Valls-SoleSolé et al., 1999*; *Carlsen et al., 2004*). These findings suggest that the RT includes some additional time that is not required for computing the upcoming response, raising the possibility that other factors might also influence the RT.

**eLife digest** Often, we need to make split-second decisions, be it to avoid an accident, outwit someone in an argument or to win a game. The time that it takes to respond to a signal, i.e., the reaction time, might be the crucial factor to help us succeed or even survive. Many people assume that the reaction time represents the time needed to prepare an action, and to respond sooner one must 'think faster'.

However, what really happens in the brain is not well understood. While more complex tasks seem to require longer reaction times, recent evidence suggests that determining when an action begins may not depend on how long it takes to decide which specific action should be taken. Indeed, reaction times may be shortened without changing the accuracy of the planned movement.

Using different performance tests, Wong et al. now demonstrate that the reaction time can be influenced by prior experience. In the first task, participants had to respond quickly to catch a moving target. When they later had to move toward a static target, their reaction times were reduced. In the second experiment, the participants practiced a task that required them to plan movements around obstacles. Participants were then given a hint that made it easier to plan their movements, but reaction times did not decrease as expected. Wong et al. then analyzed their movements and demonstrated that although reaction times remained the same, the hint did ease movement planning.

This suggests that the reaction time did not always reflect how long it took to prepare a response, but was influenced by prior experience. A next step will be to test what other factors may influence the reaction time. A deeper knowledge of these factors will help to avoid misinterpretation of neural data.

DOI: https://doi.org/10.7554/eLife.28075.002

An additional clue that the RT does not strictly represent computation time comes from evidence that the RT is influenced by context. This was illustrated in a recent study examining the planning of intentionally curved reaches (*Wong et al., 2016*). In this study, the RTs of simple point-to-point reaches were observed to be shorter than those of more complex curved movements, consistent with the idea that RT reflects computation time. However, when point-to-point movements were interleaved among curved reaches, the RTs of those point-to-point reaches were surprisingly prolonged by ~90 ms, matching the RTs of the curved reaches. A similar contextual effect has been observed during mental rotation: during a letter discrimination task, identifying letters that were presented upright occurred at much shorter RTs when all the letters were upright compared to when other letters in the same block of trials were rotated (*Ilan and Miller, 1994*). These data suggest that RTs might be subject to experience-dependent biases – akin to other movement parameters such as speed or direction (*Diedrichsen et al., 2010*; *Verstynen and Sabes, 2011*; *Hammerbeck et al., 2014*; *Huang et al., 2011*) – rather than arising strictly from the outcome of computational requirements. That is, RTs may be subject to habit in the sense that an RT of a given magnitude may become more likely to be generated in the future simply because it has been generated in the past, regardless of current task requirements.

Here we present results from two experiments which demonstrate that the RT is subject to experience-dependent effects. In the first experiment – a target-interception task – we showed that experience with initiating reaches at a short or long RT exerts a corresponding bias on the RTs of subsequently performed point-to-point reaches. In a second experiment, we demonstrated that these RT biases can also occur for more complex curved reaches around barriers. This barrier task can be performed with or without the presence of a direct cue illustrating the appropriate trajectory, corresponding to greater or lesser computational requirements respectively (*Wong et al., 2016*). We found that participants' RTs in the cued condition depended strongly on whether or not they had previously experienced the uncued condition. Analysis of the kinematics of these curved reaches allowed us to determine that the observed experience-dependent changes in RT were not due to habitually ignoring the cue and planning the movements in a different manner, but instead to habitually adopting a longer RT despite using a briefer computation (i.e., taking advantage of the cue) to solve the task. Together, these two experiments reveal that the RT does not strictly reflect the time

needed to complete the computations required for preparing responses, but may instead be selected habitually according to prior experience.

## Results

### Experiment 1: RTs were biased according to performance in a previous task

In Experiment 1, we tested whether performance of a task that encouraged movements to be generated with particular RTs could affect the RTs of subsequent movements performed in a different context. We asked participants to perform an interception task to hit a target that moved in a straight line toward or away from the participant (*Figure 1A*). For one group of 10 participants, the target always moved outward, encouraging participants to initiate reaches at shorter RTs to intercept the target before it moved beyond an invisible boundary and disappeared (see Materials and methods). For a second group of 10 participants, the target always moved inward. This encouraged participants to increase their RTs, since they could reduce the effort required to complete this task by waiting for the target to move closer before reaching out to hit it. In both cases, participants were given no specific RT instruction. Additionally, because participants were not required to stop inside the target on interception trials, participants typically generated shooting movements through the target. Immediately before and after participants performed this interception task, we measured their RTs for simple point-to-point reaches (wherein the hand had to stop inside the target and move at a constrained speed) to assess any experience-related changes in RT.

At baseline, participants in both groups generated point-to-point reaches with comparable RTs (*Table 1*; no significant difference between groups: $t = -0.22$, p=0.82), and were able to satisfy the speed requirements of the task (see *Table 1* and Materials and methods).

Movement of the target, either toward or away from the participant's starting position, exerted a reliable influence on behavior during performance of the interception task. When the target moved outward (*Figure 1B*; *Figure 1—source data 1* – Outward; *Table 1*), participants increased their movement speed (paired t-test comparing pre-training to the last training block, $t(9) = 6.30$, p<0.001) and decreased their RT (paired t-test comparing pre-training to the last training block, $t(9) = 2.99$, p=0.02) relative to their behavior on baseline point-to-point reaches. These reductions in RT persisted in the final phase of the experiment when the target was again stationary: RTs during the post-training point-to-point reaching block were shorter than in the analogous pre-training block by $21.68 \pm 6.32$ ms (*Figure 1C*; paired t-test comparing post to pre, $t(9) = -3.43$, p=0.02). This change in RT from pre-training to post-training was on average 76.6% of the total reduction in RT associated

**Table 1.** Movement parameters for Experiment 1.

| | Reaction time (ms) | Peak velocity (m/s) | Endpoint error (cm) |
|---|---|---|---|
| | Outward Interception | | |
| Pre-training | 295.38 ± 14.98 | 0.66 ± 0.02 | 1.16 ± 0.10 |
| Last training block | 255.20 ± 7.85 | 1.28 ± 0.10 | n/a |
| Post-training | 273.20 ± 12.10 | 0.67 ± 0.01 | 1.32 ± 0.22 |
| | Inward Interception | | |
| Pre-training | 290.76 ± 18.35 | 0.55 ± 0.02 | 1.09 ± 0.06 |
| Last training block | 564.82 ± 112.11 | 0.27 ± 0.04 | n/a |
| Post-training | 303.63 ± 17.25 | 0.57 ± 0.01 | 1.05 ± 0.04 |

RT, peak velocity, and endpoint error for the two groups in Experiment 1 for point-to-point movements measured before and after training, as well as for shooting movements during the last block of training. Reach velocity was required to be between 0.6 and 0.9 m/s for the outward-interception task and between 0.5 and 0.8 m/s for the inward-interception task. Note that no endpoint error is reported for the last block of training because participants reached past a continuously moving target, making it difficult to define error when participants missed the target during those blocks.

DOI: https://doi.org/10.7554/eLife.28075.003

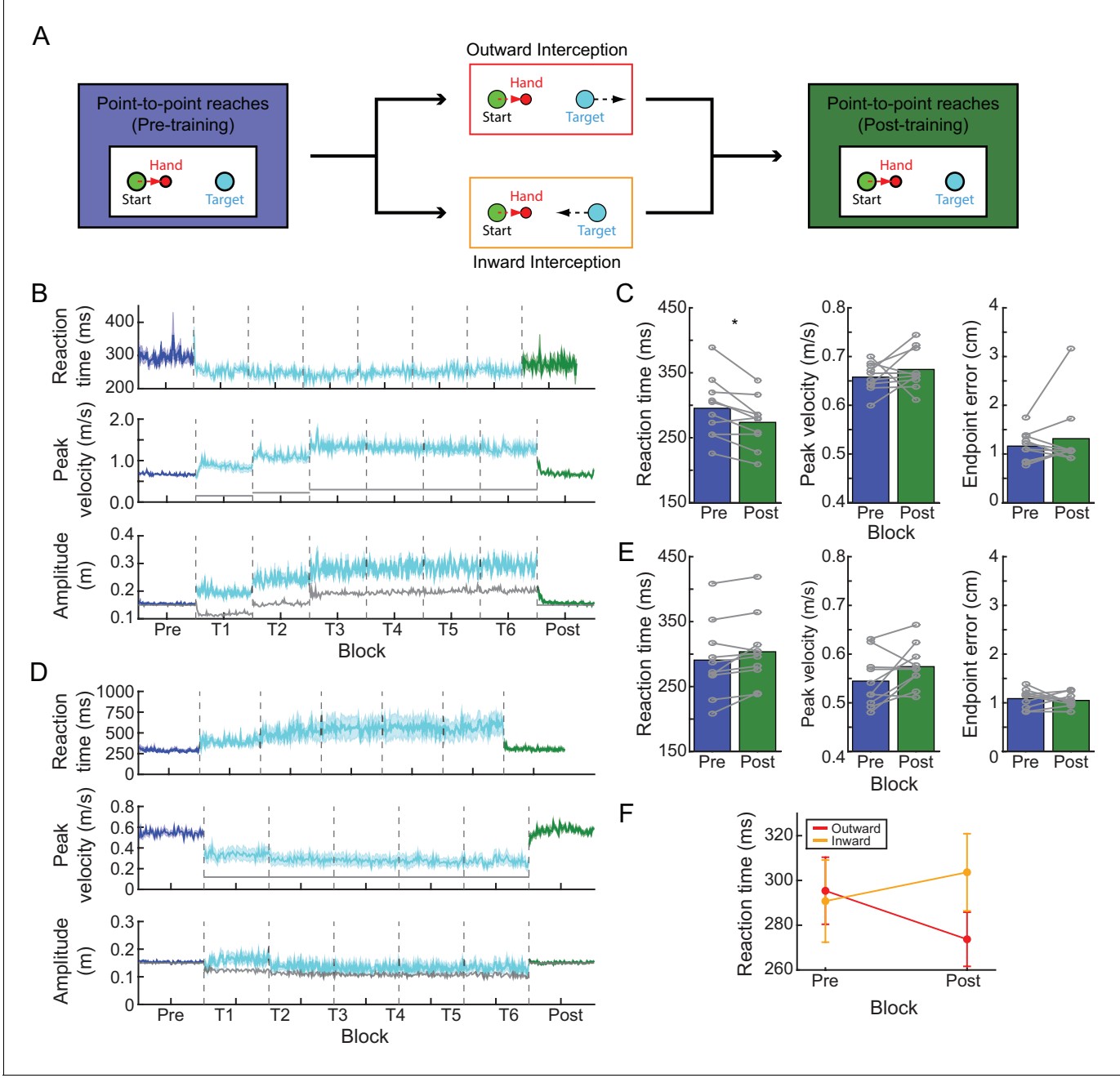

**Figure 1.** Experiment 1: interception task. (**A**) Paradigm. Participants were first asked to perform a block of point-to-point reaches by moving a cursor representing their hand (red circle; red dashed arrow reflects direction of hand movement) to hit stationary targets (cyan circle). Following this, participants in the outward-interception task were required to hit targets that moved away from them (target motion denoted by black dashed arrow), while participants in the inward-interception task were required to hit targets that moved toward them. Following this, all participants completed a final block of point-to-point reaches. (**B**) Time course of average RT, average peak velocity, and average amplitude changes during the outward-interception task. Peak velocities are compared to the actual target velocity (solid gray line) and reach amplitude is compared to the average target amplitude at interception (solid gray line; note for pre- and post-blocks the targets remained at a fixed amplitude). For reaches made during the interception task, the reported amplitude is the magnitude of the entire movement generated (i.e., until the velocity of the hand returned to zero) regardless of when or whether the target was successfully intercepted. Shaded regions represent S.E.M. (**C**) Comparison of baseline (blue) and post-training (green) performance for average RT, average peak velocity, and average endpoint error; each gray line is an individual participant. (**D,E**) As in panels B and C, but for the inward-interception task. (**F**) Across the two tasks, there was a significant interaction between task and block for average RT; this was driven by an RT shift away from baseline behavior in opposite directions for the two groups following training.

DOI: https://doi.org/10.7554/eLife.28075.004

*Figure 1 continued on next page*

*Figure 1 continued*

The following source data is available for figure 1:

**Source data 1.** Outward.
DOI: https://doi.org/10.7554/eLife.28075.005
**Source data 2.** Inward.
DOI: https://doi.org/10.7554/eLife.28075.006

with the outward-interception task (i.e., pre-training compared to the last outward-interception training block). Importantly, these changes in RT for point-to-point movements occurred with no significant change in either peak velocity (paired t-test comparing post to pre, $t(9) = 0.98$, p=0.70) or endpoint error (paired t-test comparing post to pre, $t(9) = 0.95$, p=0.70). Thus, repeatedly generating movements at low RTs led to a reduction of the RT on subsequent point-to-point reaches with no detectable decrement in performance.

Interception of an inward-moving target had the opposite effect on behavior (*Figure 1D, E*; *Figure 1—source data 2* – Inward; *Table 1*). Relative to baseline point-to-point movements, training on the inward-interception task led to a decrease of movement speed (paired t-test comparing pre-training to the last training block, $t(9) = -6.82$, p<0.001) and an increase in RT (paired t-test comparing pre-training to the last training block, $t(9) = 2.45$, p=0.04). RTs during a subsequent block of point-to-point reaches increased on average by $12.87 \pm 4.94$ ms relative to baseline, although this change was not significant (paired t-test comparing post to pre, $t(9) = 2.60$, p=0.09) and corresponded to only 9.69% of the increase in RT observed during the interception task. There was also a trend for the velocity of these reaches to increase although this also was not significant (paired t-test comparing post to pre, $t(9) = 2.07$, p=0.14) and there was no clear change in endpoint error (paired t-test comparing post to pre, $t(9) = -0.56$, p=0.59).

A mixed-effects model comparing the change in point-to-point RTs across the two groups revealed a significant interaction (likelihood ratio test; $\chi^2(1)=77.55$, p<0.001) between the direction of target motion during training (outward or inward) and the change in RT from pre-training to post-training blocks (*Figure 1F*). This interaction was driven by a difference in RT only during the post-training block (post-hoc test, p=0.03), since RTs during the pre-training block were comparable. Together, these data reveal that the RT can be biased in an experience-dependent manner, suggesting that the RT does not strictly reflect obligate computation time.

## Experiment 2: RT biases occurred despite task-dependent changes in movement planning

To further explore the experience-dependent bias in RT, we examined this effect in a more complex task that required participants to plan curved reaches around barriers. This task can be performed in one of two ways: either in the absence or presence of cues that illustrate a specific path around the barriers (*Figure 2A*). Although the movements required to complete the task are identical in both cases, the presence of a path cue has been observed to provide a significant RT advantage (*Wong et al., 2016*). This advantage is thought to reflect a difference in computational requirements between the cued and uncued conditions, with the latter condition requiring an additional trajectory-planning stage to represent the path shape to be executed. Hence, Experiment 2 provides an assay of habitual effects on RT between two sets of movements that have different planning requirements but the same execution requirements.

### Experience-dependent RT biases were observed for curved reaches around barriers

In Experiment 2A, two groups of 12 participants were asked to generate reaches around barriers in the presence or absence of path cues. Comparable to our previous findings (*Wong et al., 2016*), we observed that participants provided with a path cue exhibited a large RT advantage of 52.94 ms compared to individuals performing the task without the path cue (initial cueing condition experienced by each group; uncued: $349.76 \pm 8.81$ ms; cued: $296.82 \pm 7.68$ ms; main effect of condition, $\chi^2(1)=500.61$, p<0.001; *Figure 2C*; *Figure 2—source data 1*). Although the RT difference is smaller than observed previously (*Wong et al., 2016*), this is likely attributable to the fact that there were

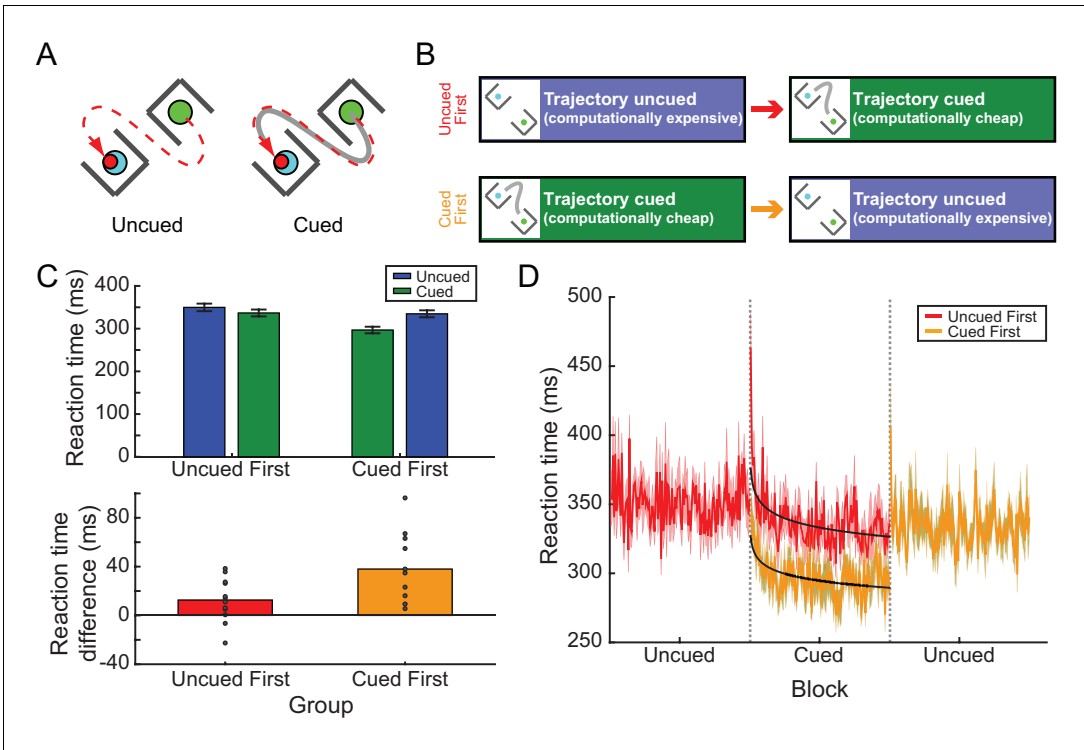

**Figure 2.** Experiment 2A: RTs depended on prior experience in the barrier task. (**A**) Participants reached from a start position to a target without intercepting a pair of barriers. One example target and barrier configuration is shown at left. Under the path-cued condition (at right), participants were also provided with a cue (solid gray line) indicating how they should get to the target. Note, in both cases the barriers, target, and cue (when present) disappeared upon movement onset; hence participants in the cued condition were not merely tracing the cue but were required to hold the desired movement trajectory in memory, analogous to individuals performing the uncued condition. (**B**) Participants were divided into two groups; one group performed an uncued reaching block followed by a cued reaching block, while the other group performed these blocks in the opposite order. (**C**) Upper panel, the average RT across all participants for each condition (cued or uncued) for the two groups (uncued-first or cued-first). Lower panel, the average difference in RT between cued and uncued reaches across all barrier configurations; positive differences indicate that RTs were shorter on cued reaches compared to uncued reaches. Dots represent individual participants. (**D**) The time course of RTs across all trials, for each group (red, uncued-first group; orange; cued-first group). Black lines represent power-law fits to the decay of RT during the cued blocks of trials. Data in the cued-first group have been horizontally offset to align the cued blocks from each group for direct comparison.

DOI: https://doi.org/10.7554/eLife.28075.007

The following source data is available for figure 2:

**Source data 1.** This file contains RT data for Experiment 2A, used to generate *Figure 2C* and *Figure 2D*.
DOI: https://doi.org/10.7554/eLife.28075.008

fewer possible barrier configurations presented in the current task compared to the previous version, that is, consistent with Hick's law (*Hick, 1952*).

Each group was then exposed to a block of trials with the opposite cue condition (*Figure 2B*); that is, participants who first performed cued trials were next asked to perform the task without path cues, and vice versa. This switch revealed a similar experience-dependent effect as in Experiment 1. Specifically, whereas there was an RT advantage associated with the availability of a cue during the initial training block, when the conditions were switched the two groups exhibited similar RTs regardless of the presence or absence of the cue (cueing condition during the second block: uncued: $334.75 \pm 8.17$ ms, cued: $336.74 \pm 8.01$; $\chi^2(1)=0.035$, p=0.85; *Figure 2C*). This experience-dependent effect of path cues on the RT was confirmed by a significant main effect of the order in which the cued and uncued conditions were experienced ($\chi^2(1)=11.55$, p<0.001), as well as a

significant interaction between condition and order ($\chi^2$(1)=105.42, p<0.001). This effect could not be attributed to changes in the RTs of uncued trials performed first compared to second (p=0.25); instead, this interaction arose because RTs were significantly longer on cued trials when participants had previously performed a block of uncued trials (p<0.001). Hence participants actually exhibited longer RTs on less computationally demanding cued trials if they experienced this condition second, even though they had more experience performing a more challenging version of the task.

Interestingly, in Experiment 2A the RTs in the cued blocks (but not the uncued blocks) exhibited a slow asymptotic decline across trials (*Figure 2D*). The rate of decline was not different between the two groups (non-significant interaction between group and trial, $t = -1.46$, p=0.14). Power-law fits to the time series of latencies yielded similar exponents for both groups (cued trials experienced first: $-0.025$; cued trials experienced second: $-0.029$). This decline suggests that both groups were responding to the appearance of the path cues in a similar manner, perhaps by learning to take advantage of the cues when planning their movements (see the section below). Nevertheless, a constant RT offset remained across the entire block, reflecting a strong, persistent RT bias for the group that had prior experience with the uncued condition ($t = 8.56$, p<0.001). Hence, even in this more complex task of generating reaches around barriers, RTs could be biased by prior experience.

In contrast to RT, task success in hitting the target while avoiding the barriers did not significantly vary with cueing condition or condition order (no effect of condition, $\chi^2$(1)=2.26, p=0.13; no effect of order, $\chi^2$(1)=2.50, p=0.11; non-significant interaction, $\chi^2$(1)=1.63, p=0.20). That is, participants who were exposed to the uncued condition first did not become more successful when they switched to the simpler, cued version of the task (uncued block: 81.97 ± 2.56% successful; cued block: 85.81 ± 1.62% successful; posthoc test, p=0.14); nor did participants who were first exposed to the cued condition become less successful when switched to the more difficult uncued task (cued block: 88.41 ± 1.66% successful; cued block: 88.02 ± 2.73% successful; posthoc test, p=1.00). Note that in all cases participants could have further improved their task success. Thus while participants who initially performed the uncued condition exhibited prolonged RTs in the subsequent cued condition, this RT increase did not confer any performance advantage (as measured by task success) as might be expected by a speed-accuracy trade-off (*Fitts, 1966*).

We observed that RTs for cued trials were longer if participants had previously performed uncued trials, but not vice versa; RTs for uncued trials were independent of whether or not participants had previously performed cued trials. This asymmetry may have arisen because participants already had the minimum possible RTs for the uncued condition, but were free to increase their RTs in the cued condition. However, it is also possible that this asymmetry was simply due to participants in the uncued-first group happening to require longer RTs even when a cue was present. To address this issue, a separate group of 12 participants (Experiment 2B; *Figure 3*; *Figure 3—source data 1*) were first exposed to cued trials, which they completed at low RTs. We then asked these participants to complete two additional blocks: a block of uncued trials, followed by a second block of cued trials. We predicted that participants would increase their RTs during the first transition (cued to uncued) but, due to the asymmetric habit effect, would not return to their previously-expressed low RTs when the cues were reintroduced.

Consistent with this prediction, participants in Experiment 2B initially generated cued reaches at low RTs (pre-exposure cued block: 349.81 ± 36.03 ms); however, experience with performing a block of uncued trials (uncued-block RT: 390.26 ± 66.37 ms) led to prolonged RTs in the second cued block (post-exposure cued block: 375.09 ± 41.38 ms; effect of block, $\chi^2$(2)=532.46, p<0.001; significant differences between all pairs of blocks, p<0.001; *Figure 3*). This increase in RT from the pre-exposure to the post-exposure cued blocks reflected an average RT bias of 80.98% of the change in the RT from the initial cued to the uncued block (average change could not be calculated for one participant because that individual exhibited no change between the initial cued block and the uncued block). Thus, RT is surprisingly biased by prior exposure to the uncued condition in this task even when participants had previously completed the cued task at low RT.

## Path cues consistently influenced how movements were planned, independent of prior experience

Returning to Experiment 2A, we next asked what was the source of the RT bias. One possibility is that participants might simply have adopted habitually long RTs regardless of the underlying

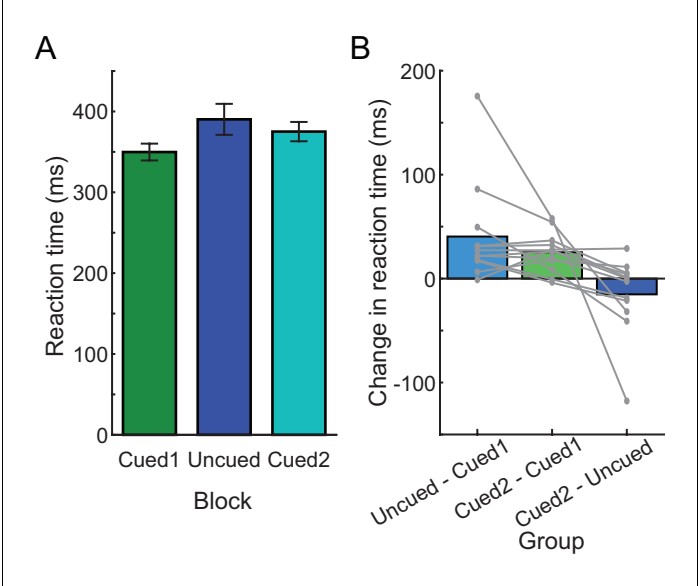

**Figure 3.** RTs in Experiment 2B. (**A**) Average RTs during each block, in the order that blocks were experienced in the experiment (from left to right). (**B**) Change in average RT between each pair of blocks during the experiment; each gray line is an individual participant.

DOI: https://doi.org/10.7554/eLife.28075.009

The following source data is available for figure 3:

**Source data 1.** This file contains RT data from Experiment 2B, used to generate *Figure 3*.
DOI: https://doi.org/10.7554/eLife.28075.010

computational demands of the current trial. Alternatively, participants that had initially been exposed to uncued trials might have ignored the cues when they became available and instead continued to plan the movement trajectories de novo – a more computationally demanding means of planning that requires prolonged RTs (*Wong et al., 2016*). To distinguish between these possibilities, we examined the kinematics of movements made with and without path cues, to find evidence of qualitatively different modes of movement planning. Specifically, we reasoned that the trajectories of movements planned using the cue should more closely resemble the path cues than movements planned de novo, regardless of the RTs at which these movements were initiated.

We found that the presence of path cues significantly influenced movement kinematics in a consistent manner across participants: the shapes of the average trajectories for the cued and uncued conditions (pairs of blue and green lines in *Figure 4A*) were significantly different for at least one point along the length of the trajectory in 70% of all the target-barrier configurations performed (*Figure 4A*, significant differences between trajectory pairs along the movement are denoted by yellow dots). For those trajectories where differences were observed, on average 21% of the movement was found to be significantly different depending on the condition. These differences often occurred near the start of the movement, wherein the hand passed closer to the starting barrier on cued trials. However, there was no significant effect of the order in which the conditions were performed; moreover, no kinematic differences were noted when comparing reaches performed in the same cueing condition (i.e., between cued trials experienced first or second, or between uncued trials experienced first or second). Thus cued reaches were always executed in a similar fashion regardless of whether participants had prior experience with the uncued condition.

In fact, a Procrustes shape-comparison analysis (*Goodall, 1991*; *Figure 4B*; *Figure 4—source data 1* – Procrustes Distance) revealed that reaches significantly resembled the path cue more closely when it was available (effect of condition: −0.13, 95% confidence interval: [−0.19,–0.08], Bayes factor = 0). In contrast, there was no significant effect of order on trajectory shape (effect of order: −0.04 [-0.1, 0.02], Bayes factor = 14.24), even when comparing only cued trials (effect of order: 0.04 [-0.02, 0.09], Bayes factor = 13.63). Together, these data suggest that participants used

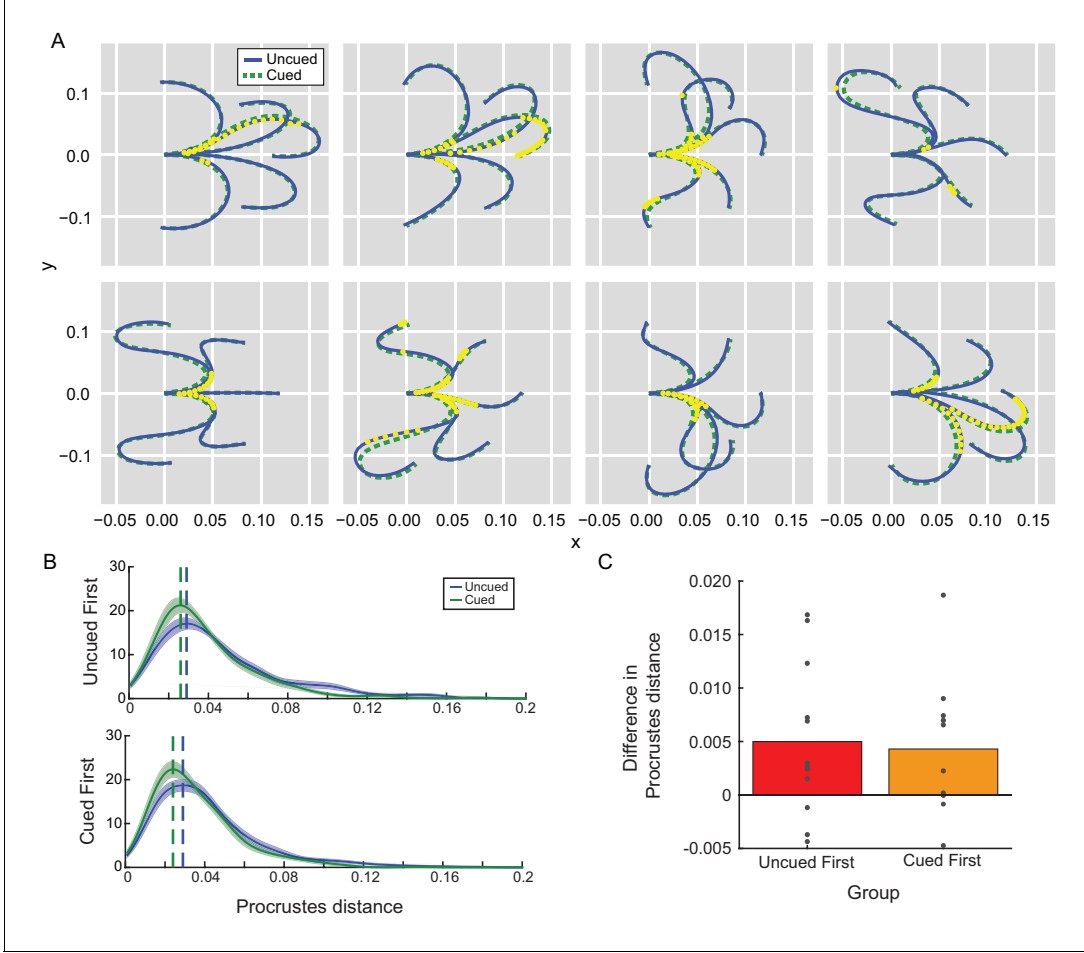

**Figure 4.** Movement kinematics in Experiment 2A depended on the presence of the path cue. (**A**) Average reach trajectories across participants for each barrier configuration for uncued (blue) and cued (green) conditions. Each yellow dot represents a time at which the mean trajectories were found to be consistently different between conditions. Since no significant effect of order was observed, data were collapsed across groups. (**B**) Trajectory shapes for cued and uncued reaches were examined for their similarity to the path cue according to a Procrustes distance analysis. A Procrustes distance of zero means the trajectory is identical in shape to the path cue. Distributions of estimated Procrustes distances pooled across all trajectory shapes were averaged across all participants in each group (uncued-first or cued-first) and are shown for comparison; the mode of each distribution is indicated by the vertical dashed line for visualization purposes. No significant effect of order (group) was observed. (**C**) Average difference in Procrustes difference between uncued and cued reaches for each individual participant is shown; positive differences indicate that cued reaches are more similar in shape to the path cue compared to uncued reaches.

DOI: https://doi.org/10.7554/eLife.28075.011

The following source data is available for figure 4:

**Source data 1.** Procrustes Distance.

DOI: https://doi.org/10.7554/eLife.28075.012

the path cue to simplify planning their movements whenever it was available (*Wong et al., 2016*); that is, participants were changing how they planned their movements in response to the cue, which resulted in quantitative changes in the kinematics of their actions such that movements more closely resembled the cued paths. Similar effects were observed in Experiment 2B (*Supplementary file 1* for Experiment 2B kinematics analysis); in fact, the kinematics in the second cued block resembled the path cues more closely than even those of the initial cued block (effect of block: −0.04, 95% confidence intervals [−0.07,–0.02], Bayes factor = 0.39) – a finding that would be expected to reduce RTs as participants took advantage of the cue, not prolong them (e.g., consistent with Experiment 2A). Hence despite evidence suggesting that reaches were planned on a trial-by-trial basis in an experience-independent manner (in particular by relying on the cue to plan the action), RTs did not

consistently reflect the computational benefit afforded by the path cue. Instead, RTs exhibited an experience-dependent bias that was unrelated to how these reaches were being planned and executed.

## Discussion

Although the RT is typically assumed to directly reflect the computation time required to plan a movement (*Donders, 1969*; *Sternberg, 1969*; *Friston et al., 1996*; *Spivey, 2007*; *Sanders, 1998*), we demonstrated here that the RT may instead be influenced by previous experience. This was not simply a practice effect leading to more efficient planning of the same action; RT was influenced by recently generated RTs in the past even when those RTs were produced during another task with a different type of movement (e.g., in Experiment 1 the RTs of shooting movements to intercept a target biased the RTs of subsequent point-to-point reaches to stationary targets). This effect, while small, is comparable in magnitude to the RT changes previously observed in response to the addition of a delay period between stimulus and go cues (*Churchland et al., 2006*), or to changes in stimulus or response complexity (*Fitts, 1954*; *Fitts, 1966*; *Hick, 1952*; *Simon, 1967*; *Henry and Rogers, 1960*, for review, see *Teichner and Krebs, 1974*). Moreover, we showed that RTs can be modulated to be either shorter or longer than previously expressed depending on the nature of the interposed task, and that these effects persisted across a large number of subsequent trials. Thus in general, the RT expressed on any given trial may be lengthened or shortened (by 20–30 ms) strictly due to prior experience, with no other measurable impact on performance.

### Changes in motivation cannot explain RT biases

Experiment 1 provides clear evidence that the RT can be modulated in an experience-dependent manner as if by a habit, and that such biases persist over a large number of trials. However, this paradigm was unable to ascertain whether these RT biases arose because of changes in the rate of computational processing, or because of effects on the non-computational portion of the RT. In the former case, changing the duration of computational processes could increase task success, particularly when shortening the RT in the outward-interception task. Thus there was a motivational incentive that could drive changes in the speed-accuracy trade-off in favor of shortened RTs while maintaining consistent accuracy (e.g., motivation could have driven an increase in the rate at which evidence accumulated in a drift-diffusion model of RT [*Ratcliff, 1978*]), much in the same way that reward can reduce RT without affecting accuracy (*Takikawa et al., 2002*; *Hübner and Schlösser, 2010*; *Salinas et al., 2014*; *Manohar et al., 2015*). This modified speed-accuracy trade-off may then have been retained when planning subsequent point-to-point movements. Alternatively, if RT biases simply reflect modulation of the non-computational portion of the RT, the manner in which movements are planned should remain unchanged.

Two pieces of evidence from Experiment 1 speak against the hypothesis that the observed changes in RT occurred through a motivational effect. First, the RT bias persisted across a large number of trials (e.g., the entire next block) with no obvious decay toward the initially expressed RT. This persistence stands in contrast to the short-lived effects of motivation, which appear to decay after only a few subsequent trials (*Takikawa et al., 2002*; *Xu-Wilson et al., 2009*; *Wong et al., 2015*). Second, the effect of motivation typically influences not just the RT, but also movement speed and accuracy; in general, these three parameters have often been observed to modulate together in response to motivation (*Takikawa et al., 2002*; *Wong et al., 2015*). However, we observed no obvious changes in other movement kinematics such as movement speed or accuracy that accompanied the RT biases. On the other hand, speed and accuracy were constrained by task requirements during the point-to-point movement blocks. Nevertheless, these data suggest that motivation alone cannot account for the long-lasting RT biases observed in Experiment 1.

### Habitual initiation rather than habitual planning

The kinematically more complex trajectories required in Experiment 2 provided us with greater sensitivity to examine whether RT biases arose from effects on the computational or the non-computational portion of the RT. That is, we were better able to detect any execution-related differences that may have arisen from subtle changes in movement planning that were associated with changes in RT.

During Experiment 2, participants in the cued condition could have initiated their movements at short RTs; however, they exhibited persistently prolonged RTs when previously exposed to the more computationally demanding uncued condition. Such a bias in RT could have occurred for one of two reasons. Participants may have persisted in planning trajectories in the same manner as they had done without path cues, despite the availability of the path cue that would eliminate the need for this planning stage (*habitual planning*). Alternatively, participants may have used the path cue to reduce the computational load for planning but simply did not express their prepared actions earlier; that is, perhaps the RT does not strictly reflect computation time, but instead contains a manipulable, non-computational component that may be subject to habit (*habitual initiation*).

The analysis of movement kinematics allowed us to distinguish between these two alternatives. In particular, we demonstrated that movement kinematics changed in a consistent manner in the presence or absence of the path cues, arguing that differences in movement kinematics do reflect changes in how these actions were being planned. However, we observed no effect of condition order on trajectory kinematics (i.e., reach kinematics were always influenced similarly by the path cues when they were available, regardless of whether participants had prior experience with uncued trials), suggesting that participants did not ever exhibit habitual cue-free preparation of the movement trajectory when generating reaches in the presence of a cue. This is particularly evident in Experiment 2B, when movement kinematics indicate a particularly strong reliance on the cue to plan the action in the second cued block even though the RT did not reflect such changes in movement planning. Moreover, reaches in the uncued and cued conditions were confined to different portions of the workspace, making it unlikely that participants habitually applied the identical motor plans from previously-performed uncued reaches. These findings stand in contrast to the RT, which did not consistently modulate according to the presence or absence of the path cue, but was instead experience-dependent. Hence these data suggest that unlike motor planning – which is determined by the current task at hand – movement onset may be influenced in a use-dependent manner consistent with the idea of habitual initiation, and therefore may not represent the actual time required to prepare a movement.

If the RT can be habitual, this implies that the time of movement initiation is not simply the end result of computational processing but may actually be represented as a separate movement parameter during the pre-movement period, analogous to movement speed. Indeed, previous work (*Haith et al., 2016*; *Brown and Robbins, 1991*) has suggested that movement initiation may be independent of movement planning. Such a distinction between planning and initiation is consistent with recent data suggesting that an initiation signature can be observed in motor cortex preceding movement onset in a manner that is independent of the specific action being prepared (*Kaufman et al., 2016*). Our finding that movement initiation can be determined by prior experience, rather than the time when planning has completed, further supports this view.

Selection of RT based on prior experience rather than computation time implies that it should be possible to choose a RT that is too short to allow for all planning processes to be completed prior to movement initiation. Consistent with this idea, there indeed appear to be situations in which the RT is spontaneously chosen to be improperly short (*Orban de Xivry et al., 2017*; *Haith et al., 2016*). In such cases, however, online corrections and refinements that occur after movement initiation help ensure a successful movement outcome (*Orban de Xivry et al., 2017*; *Kohen et al., 2017*; *Wong and Haith, 2017*). Hence, in the event that the RT is biased to be shorter than the time required to complete all necessary decision-making and planning-related computations, the motor system has online correction mechanisms in place to maintain overall task success.

## A common feature of movement parameters is that they are subject to experience-dependent biases

Experience-dependent biases have been previously demonstrated for movement parameters such as speed or direction. For example, repetition of movements at a particular speed or toward a particular direction of the workspace strongly influences the kinematics of subsequently performed reaches (*Diedrichsen et al., 2010*; *Verstynen and Sabes, 2011*; *Hammerbeck et al., 2014*; *Huang et al., 2011*). Repetition has also been shown to influence the direction of movement invoked by transcranial magnetic stimulation (*Classen et al., 1998*). In these cases, however, the influence of prior experience on movement biases appears for the most part to be short-lived, unlike the long-lasting effects seen here in the RT.

Why experience-dependent biases occur is still unclear. It has previously been proposed that such biases may simply be a result of Hebbian learning (*Bütefisch et al., 2000*; *Bütefisch et al., 2004*): repetition of the same action presumably strengthens the neural circuits that give rise to that movement, making it more likely to be invoked in the future. From a theoretical standpoint, experience-dependent biases have been framed in the context of a normative Bayesian model in which repetition leads to the construction of a strong prior that influences the preparation of future responses (*Verstynen and Sabes, 2011*). Regardless of whether one uses a mechanistic or computational framework, however, explanations for experience-dependent biases rely on the assumption that the parameter in question – e.g., speed or direction – is represented as a movement parameter that can be specified prior to movement. Therefore, experience-dependent biases on the RT imply that the RT is not simply the passive consequence of the time required to complete computational processing, but may instead be represented as a separate movement parameter that can be subject to habit.

## Conclusions

In summary, these data support the hypothesis that the RT should be considered a distinct parameter that is selected during the pre-movement period; that is, the RT may in part reflect, but is not strictly dependent upon, underlying computational processes. Thus, caution must be taken when interpreting differences in RT as indicative of the existence of computational stages with smaller or larger processing demands. Such RT differences may simply reflect carryover from RTs recently exhibited in the past (i.e., habit), regardless of computational demands.

# Materials and methods

Fifty-six right-handed, adult (average age, 23.27 years old; 26 males) neurologically healthy participants were recruited for this study. Twenty individuals participated in Experiment 1 (10 individuals in each group); 24 individuals participated in Experiment 2A (12 individuals per group), and 12 individuals participated in Experiment 2B. Group sizes were chosen based on findings from previous studies (*Wong et al., 2016*). All participants provided written informed consent and were naive to the purposes of the study. Experimental methods were approved by the Johns Hopkins University School of Medicine institutional review board.

Participants made planar reaching movements with their right arm along the surface of a glass table. Their wrist was restrained in a wrist splint, and supported by pressurized air jets to allow frictionless movement of the elbow and shoulder. Vision of the arm was obscured by a mirror through which participants observed an LCD monitor (60 Hz), which displayed targets and a cursor representing the position of the index finger in a veridical horizontal plane. Movement of the index finger was tracked using a Flock of Birds magnetic tracking system (Ascension Technology, VT, USA) at 130 Hz.

## Experimental paradigms

### Experiment 1: Interception task

Targets appeared at any of 4 possible positions uniformly spaced every 90° along a circle centered on the start position. After a brief delay of 100 ms, the target immediately began moving, and the participant was required to intercept the target before it vanished (*Figure 1A*). In the outward-interception task, the target appeared 4 cm from the start position and moved outward until it was either intercepted or it reached 26 cm away from the start position, at which point it disappeared. Blocks consisted of 60 reaches (15 to each of the 4 target directions presented in a pseudo-randomized order); the target moved at a speed of 0.15 m/s in the first block, 0.225 m/s in the second block, and 0.3 m/s for the remaining 4 blocks. In the inward-interception task, the target appeared 24 cm away from the subject and began moving inward at a velocity of 0.12 m/s during each of the 5 blocks of 60 trials until it was intercepted or it reached the starting position. Participants had only one attempt to hit the target; the trial ended when the hand stopped moving (velocity <0.05 m/s) or the hand exceeded the current target radius from the starting position.

Prior to and following these interception blocks, participants completed 60-trial blocks in which they were required to make point-to-point reaching movements to stationary targets located 15 cm away from the start position, uniformly spaced every 90°. Participants were also encouraged to

satisfy a velocity criterion on each trial (between 0.6 and 0.9 m/s for the outward interception task, or between 0.5 and 0.8 m/s for the inward interception task), although no trials were excluded from analysis based on movement velocity. These differences in velocity criterion were deliberately chosen to encourage reaches at speeds that were comparable to those made during the interception training. Encouraged adherence to a velocity criterion simply allowed for an assay of changes in RT due to training on the interception task independently of any potential changes in movement speed. For both the pre- and post-training blocks, the first 10 trials were discarded from analysis to ensure that block averages for each participant reflected steady-state behavior.

## Experiment 2: Barrier task

Experimental methods were similar to a barrier-avoidance task reported previously (*Wong et al., 2016*) in which participants were required to generate a reaching movement toward a target while avoiding barriers that appeared around both the start position and the goal target (*Figure 2A*). Targets appeared at any of eight positions uniformly spaced every 45° along a circle of radius 12 cm; barriers consisted of three-sided boxes that could be oriented at any of eight angles (rotated in 45° increments). Participants initiated a trial by moving their hand into a central start position; after a random delay (600–1600 ms), the goal target and two barriers appeared (one around the start position and one around the goal target). Participants were instructed to begin their movement as soon as possible, and to reach toward the goal target while avoiding the barriers. As soon as participants initiated their reach, all visual information except the start position and the hand cursor disappeared. Participants stopped moving when they thought they reached the target location; upon conclusion of the movement they received visual feedback about their movement trajectory. In Experiment 2A, participants completed two blocks of 128 trials; in Experiment 2B, participants additionally completed a third block of 128 trials.

During one of the two blocks (or for Experiment 2B, the first and third blocks), participants were given a path cue in the form of a line drawn between the start position and goal target that navigated around the barriers. Additionally, in each block targets were confined to either the upper left or the lower right half of the screen such that all movements for a given block were confined to specific portions of the workspace. The direction of targets was different in the first and second block; thus, participants never performed the identical movements with and without a path cue, although cued and uncued reaches were rotationally symmetric (affording kinematic comparisons). The target direction associated with a given block was counterbalanced across participants (in Experiment 2B the target direction was the same for the first and third blocks for any given participant).

The presented path cues represented the average hand path taken to avoid the barriers from a prior experiment (*Wong et al., 2016*). The participants in Experiment 2A were subdivided evenly into two groups; in one group, participants performed uncued trials first to establish a history of performing a computationally difficult task, followed by cued blocks of trials (Group 1: uncued trials first, cued trials second) while the remaining participants were given the opposite order of blocks (Group 2: cued trials first, uncued trials second). This allowed us to examine whether experience at performing reaches in the presence or absence of a path cue could bias future RTs. In Experiment 2B, all participants performed a block of cued trials, a block of uncued trials, and a final block of cued trials. Note that for both Experiment 2A and 2B, an equal number of participants within each group experienced cued reaches to the upper left as experienced cued reaches to the lower right; thus, kinematic differences across subjects could not be attributed to any movement direction-specific biases.

## Data analysis

Data were analyzed offline using programs written in MATLAB (The MathWorks, Natick, MA) and in R (*R Development Core Team, 2016*). Analysis code and data are available on GitHub at https://github.com/BLAM-Lab-Projects/RT_habit (*Wong, 2017*). A copy is archived at https://github.com/elifesciences-publications/RT_habit. Reaches were selected according to a velocity criterion (tangential velocity greater than 0.05 m/s) and verified by visual inspection. For each movement, RT was computed as the time between target onset and movement initiation. Inherent delays in the system were estimated to be 105 ms on average; all RTs have been corrected to compensate for this delay. Velocity was calculated by taking the numerical derivative of the hand position after smoothing using

a second-order Savitzky-Golay filter with a frame size of 19 samples. Endpoint error was calculated as the absolute radial distance between the final position of the hand and the center of the target. All values are reported along with S.E.M.

In Experiment 1, RT, peak velocity, and interception amplitude were measured during interception training blocks. The amplitude of the target at the time of interception was calculated as the distance of the target away from the central starting position at the first time the hand entered the target; if the participant was not successful at intercepting the target, no target amplitude was recorded for that trial. During pre- and post-interception blocks, RT, peak velocity, and endpoint error were measured for point-to-point reaches. These three metrics were compared within groups using paired t-tests, with p-values adjusted for multiple comparisons using Bonferroni-Holm corrections. RT was also compared across groups using a mixed-effects model in R using the *lme4* package (*Bates et al., 2015*), with post-hoc pairwise tests performed using the generalized linear hypothesis testing function in the *multcomp* package (*Hothorn et al., 2008*) and adjusted for multiple comparisons using the Bonferroni-Holm correction.

In Experiment 2, trials were excluded if participants did not complete their reach within 1200 ms. Additionally, since most barrier configurations had more than one possible solution (e.g. above or below a barrier), we pre-selected one possible path as 'canonical' and presented that solution on path-cued trials; any 'non-canonical' reaches were excluded to allow for a fair comparison of RTs for movements of comparable kinematics. Reaches were not excluded if participants simply hit one of the barriers or did not reach close enough to the target to be considered successful on that trial but otherwise satisfied the inclusion criteria noted above. On average across participants, about 7.5% of reaches were excluded from uncued blocks and 3.0% of reaches were excluded from cued blocks.

RTs were compared in R with mixed-effects models using the *lme4* package (*Bates et al., 2015*). For Experiment 2A, this model treated order (uncued reaches first or cued reaches first) and condition (cued or uncued) as fixed effects and barrier configuration and participant as random effects. Significant effects were determined using a likelihood ratio test to compare pairs of models (with and without the factor of interest). In Experiment 2B, there was only a main effect of block (1, 2, or 3), with post hoc tests performed in R using the generalized linear hypothesis testing function in the *multcomp* package (*Hothorn et al., 2008*). All p-values obtained from post hoc tests were adjusted for multiple comparisons using Bonferroni-Holm corrections.

The time course of the change in RT during path-cued blocks was compared across groups by fitting generalized linear models in R with factors of group and trial using the *nlme* package (*Pinheiro et al., 2014*) to account for the autocorrelated covariance structure across time within participants. These models remove any autocorrelation structure across trials for each participant individually prior to examining main effects; the form of the autocorrelation structure was selected by fitting Autoregressive-Moving Average (ARMA) models to the data and selecting the model fit that yielded the lowest Aikake Information Criterion on average across participants; this led to a choice of an ARMA(1,1) process.

Movement kinematics were examined using two methods. First, movements were compared using tools of functional data analysis (*Goldsmith and Kitago, 2016*). Briefly, trajectories were time-normalized and evenly resampled. For each trajectory, mean pairwise differences were examined using a function-on-scalar regression model fit using a Bayesian method that allows for correlations in the errors. A 95% simultaneous posterior credible interval was used to identify significant differences between conditions, accounting for the multiple comparisons made across time points within a single trajectory but not for multiple comparisons across trajectories.

Second, movement trajectories were also examined by comparing the shape similarity of each movement to the path cue using a Procrustes distance metric (*Goodall, 1991*). The Procrustes distance finds the best combination of translation, rotation, and scaling to match a shape to its template, then estimates the remaining dissimilarity normalized between 0 and 1, where 0 implies the two shapes are perfectly matched. For each movement, the Procrustes distance was estimated; then the overall distributions of Procrustes distances for the cued and uncued conditions were compared using a generalized linear mixed model with a log-normal link function using *brms*, an R interface to the Stan language (*Buerkner, 2016*; *Carpenter et al., 2016*; *Hoffman and Gelman, 2014*). This generalized linear mixed model had main effects of order (uncued reaches first or cued reaches first) and condition (cued or uncued); significant effects were estimated by calculating Bayes factors to

test the null hypothesis that the effect coefficient is equal to zero (according to whether the confidence intervals of the effect included zero).

## Acknowledgements

This work was supported by NSF grant BCS-1358756 to John W. Krakauer, and NIH grants R01-NS097423 and R01-HL123407 to Jeff Goldsmith.

## Additional information

### Funding

| Funder | Grant reference number | Author |
| --- | --- | --- |
| National Science Foundation | BCS-1358756 | Adrian M Haith<br>John W Krakauer |
| National Institute of Neurological Disorders and Stroke | R01-NS097423 | Jeff Goldsmith |
| National Heart, Lung, and Blood Institute | R01-HL123407 | Jeff Goldsmith |

The funders had no role in study design, data collection and interpretation, or the decision to submit the work for publication.

### Author contributions

Aaron L Wong, Conceptualization, Resources, Data curation, Software, Formal analysis, Validation, Investigation, Visualization, Methodology, Writing—original draft, Project administration, Writing—review and editing; Jeff Goldsmith, Resources, Software, Formal analysis, Validation, Methodology, Writing—review and editing; Alexander D Forrence, Software, Formal analysis, Writing—review and editing; Adrian M Haith, Conceptualization, Supervision, Investigation, Methodology, Writing—original draft, Writing—review and editing; John W Krakauer, Conceptualization, Resources, Supervision, Funding acquisition, Investigation, Methodology, Writing—original draft, Project administration, Writing—review and editing

### Author ORCIDs

Aaron L Wong http://orcid.org/0000-0001-7211-0653
Alexander D Forrence http://orcid.org/0000-0002-9728-6337
Adrian M Haith http://orcid.org/0000-0002-5658-8654
John W Krakauer http://orcid.org/0000-0002-4316-1846

### Ethics

Human subjects: All participants provided written informed consent and were naive to the purposes of the study. Experimental methods were approved by the Johns Hopkins University School of Medicine institutional review board.

### Decision letter and Author response

Decision letter https://doi.org/10.7554/eLife.28075.015
Author response https://doi.org/10.7554/eLife.28075.016

## Additional files

### Supplementary files

• Supplementary file 1. Source data for Experiment 2B kinematics analysis. These data contain the Procrustes Distance measure (range, 0 to 1) calculated for each trajectory performed by every participant in Experiment 2B. The file includes a column for condition (cued block = 0, uncued = 1) and a column for block (first cued block = 1, uncued block = 2, second cued block = 3). These data were

used to calculate kinematic differences between blocks using a generalized linear mixed model in *brms* (see Materials and methods). There is no corresponding figure for these data, summary statistics are reported in the Results.

DOI: https://doi.org/10.7554/eLife.28075.013

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
