## [Decision Letter]

Thank you for submitting your article "Habitual selection of reaction times" for consideration by *eLife*. Your article has been reviewed by two peer reviewers, one of whom, Jennifer L Raymond (Reviewer #1), is a member of our Board of Reviewing Editors and the evaluation has been overseen by Timothy Behrens as the Senior Editor. The following individual involved in review of your submission has agreed to reveal their identity: Mark M Churchland (Reviewer #2).

The reviewers have discussed the reviews with one another and the Reviewing Editor has drafted this decision to help you prepare a revised submission.

Reaction time (RT) is one of the most widely used metrics in studies of primate behavior, based on the idea that it is an external reflection of internal neural processing time. This idea has been very influential, and had extensive impact on how neural data are interpreted. The present study challenges this view, by making a compelling case that RT is significantly affected by previous experience, and hence is not necessarily a reliable indicator of the time required for computation on the current trial. The study is simple and well designed. The results are appropriately analyzed and have broad implications for the interpretation of the widely used RT measure. The few concerns of the reviewers can be addressed with editing of the text for clarity.

Reviewer #1:

This study makes a case that reaction time (RT) is a free parameter that can be affected by past experience, rather than a simple consequence of the time required for computation in the current trial. The results have broad implications for the interpretation of the widely used RT measure. In general, the study appears to be well designed, and the results appropriately analyzed. My main reservation is that some of the effects sizes are quite small, making the results less convincing. Also, the presentation could be improved in some places to make this relatively simple study easier to digest by the reader.

In Figure 1, the increase in RT was not significant, and hence should not be described as an increase in subsection “Experiment 1: RTs were biased to be longer or shorter according to performance in a previous task”, and other places in the text.

Figure 4: differences in trajectory are small, and inconsistent between subjects.

Subsection “Experience-dependent RT biases were observed for curved reaches around barriers”: Success in hitting the target while avoiding barriers should be reported. If the success rate is already close to 100% for uncued, then a ceiling effect could be what is preventing detection of an effect on performance despite the altered reaction time.

Reviewer #2:

This is a pretty straightforward study, whose conclusions might almost be considered too simple. However, given that the RT is one of the most heavily studied aspects of human behavior, it is essential to understand the factors that influence it.

Historically, the RT has been heavily used because it is one of the few external measures of internal processing time. Without neural recordings, one cannot view internal processes such as movement planning / preparation. However, it is generally believed / assumed that one can at least use the RT to measure how long those processes take. In the simplest version of this idea, movement planning / preparation are key events whose culmination results in movement onset. The simplest version would be activity rising to a threshold. Under this conception, the time from stimulus to movement onset by definition reflects the time for movement preparation. A more nuanced (and probably more accurate) view is that preparation is independent from movement triggering, but that triggering normally waits for preparation to finish. In this view, the RT is typically still assumed to mostly reflect the time for preparatory processing to complete, because it is assumed that motivated subjects (humans or animals) pull the trigger almost immediately upon completion of preparation. The idea that the RT is a behavioral window on processing time has been very influential, and has extensively impacted how neural data are interpreted. For example, the ability to relate pre-movement neural activity to RT has been important in understanding the role of that activity, and in making the case that it is preparation-related.

The present study makes a strong case that the RT reflects enough different factors that it shouldn't necessarily be considered a reliable indicator of computational processing time. The changes in RT due to these 'non-processing-time' factors are not large, but they are as large as the processing time that is usually inferred from changes in RT with delay period (20–100ms depending on task difficulty). I was persuaded by the experiments that the RT is indeed often driven by factors other than processing time. This might seem kind of obvious, but it isn't considered that often (or if it is, it tends to be ignored as an inconvenient fact). The present work is compelling, and relates to other recent work from the same lab that argued that the time of movement onset can be chosen independently of the time for preparation to complete. The present work demonstrates that this is perhaps the normal state of affairs, with the RT reflecting a variety of history-based effects. This is in some ways inconvenient and annoying: it means that the RT should be used with caution as a metric of computation time. On the other hand, the present work opens up new avenues to studying the factors that determine when the 'movement trigger' is pulled, and how that process is or isn't influence by the progress of movement preparation. Both these contributions (the caution, and the new avenue) are important.

The manuscript is generally well written, but could be tightened and clarified in places. I found the 'Changes in motivation' section of the Discussion a bit rambling and confusing. I kept being confused regarding whether increased motivation was being proposed to be the same thing as a speeding of the computational process. I would have usually thought of these as different (e.g., I might really want to solve the problem faster, but there is a limit) but I can see others as thinking they are the same. This section read as a bit reactionary against a potential criticism I didn't really understand. The clearest part of the Discussion is definitely the conclusions, which lay out a clear set of statements regarding how the present findings should be interpreted.

---

## [Author Response]

The reviewers have discussed the reviews with one another and the Reviewing Editor has drafted this decision to help you prepare a revised submission.Reaction time (RT) is one of the most widely used metrics in studies of primate behavior, based on the idea that it is an external reflection of internal neural processing time. This idea has been very influential, and had extensive impact on how neural data are interpreted. The present study challenges this view, by making a compelling case that RT is significantly affected by previous experience, and hence is not necessarily a reliable indicator of the time required for computation on the current trial. The study is simple and well designed. The results are appropriately analyzed and have broad implications for the interpretation of the widely used RT measure. The few concerns of the reviewers can be addressed with editing of the text for clarity.
*Reviewer #1:*
*This study makes a case that reaction time (RT) is a free parameter that can be affected by past experience, rather than a simple consequence of the time required for computation in the current trial. The results have broad implications for the interpretation of the widely used RT measure. In general, the study appears to be well designed, and the results appropriately analyzed. My main reservation is that some of the effects sizes are quite small, making the results less convincing. Also, the presentation could be improved in some places to make this relatively simple study easier to digest by the reader.*

Although the effect sizes in our study are quite small, as Reviewer #2 notes they are on the order of the effects on processing time (~20-100ms) typically reported in other studies seeking to measure how RT varies given differing delay periods, increasing numbers of stimuli or responses, changes in task difficulty, with learning, and so forth. Hence, although these effects comprise only a fraction of the total RT, they have an important bearing on the interpretability of other studies that attempt to draw conclusions about processing time based on changes in RT.

*In Figure 1, the increase in RT was not significant, and hence should not be described as an increase in subsection “Experiment 1: RTs were biased to be longer or shorter according to performance in a previous task”, and other places in the text.*

We have revised the wording as appropriate.

Figure 4: differences in trajectory are small, and inconsistent between subjects.

We agree that there are inconsistencies between different movement trajectories, but this may be due to idiosyncrasies with particular curved-movement shapes themselves that makes them more or less difficult to execute, for which we did not have a strong *a priori* hypothesis. However, this figure and our findings summarize a repeated-measures analysis across participants per trajectory; thus, any significant point (i.e., yellow dot) indicates a consistent difference in the manner in which that trajectory was executed across all participants. Since these effects appear fairly small, we also performed the procrustes distance analysis to allow us to better quantify how different these trajectories were from each other in comparison to the path cue. We have now clarified this point.

*Subsection “Experience-dependent RT biases were observed for curved reaches around barriers”: Success in hitting the target while avoiding barriers should be reported. If the success rate is already close to 100% for uncued, then a ceiling effect could be what is preventing detection of an effect on performance despite the altered reaction time.*

We have added these scores. The success rate is fairly high, but is not close enough to 100% to suggest that there is a ceiling effect preventing detection of a performance difference on cued trials that would reflect the observed differences in RT. If anything, there is a bigger difference in success between groups than across conditions within group, again suggesting a ceiling effect is unlikely. In any event, our kinematic analysis is a far more sensitive measure of changes in performance than task success. While we can see a clear change in how movements were executed depending on condition, there was no effect of order (group) or interaction between group and condition. This more strongly supports the argument that modulations of RT do not impact performance.

*Reviewer #2:*
*This is a pretty straightforward study, whose conclusions might almost be considered too simple. However, given that the RT is one of the most heavily studied aspects of human behavior, it is essential to understand the factors that influence it.*
*Historically, the RT has been heavily used because it is one of the few external measures of internal processing time. Without neural recordings, one cannot view internal processes such as movement planning / preparation. However, it is generally believed / assumed that one can at least use the RT to measure how long those processes take. In the simplest version of this idea, movement planning / preparation are key events whose culmination results in movement onset. The simplest version would be activity rising to a threshold. Under this conception, the time from stimulus to movement onset by definition reflects the time for movement preparation. A more nuanced (and probably more accurate) view is that preparation is independent from movement triggering, but that triggering normally waits for preparation to finish. In this view, the RT is typically still assumed to mostly reflect the time for preparatory processing to complete, because it is assumed that motivated subjects (humans or animals) pull the trigger almost immediately upon completion of preparation. The idea that the RT is a behavioral window on processing time has been very influential, and has extensively impacted how neural data are interpreted. For example, the ability to relate pre-movement neural activity to RT has been important in understanding the role of that activity, and in making the case that it is preparation-related.*
*The present study makes a strong case that the RT reflects enough different factors that it shouldn't necessarily be considered a reliable indicator of computational processing time. The changes in RT due to these 'non-processing-time' factors are not large, but they are as large as the processing time that is usually inferred from changes in RT with delay period (20–100ms depending on task difficulty). I was persuaded by the experiments that the RT is indeed often driven by factors other than processing time. This might seem kind of obvious, but it isn't considered that often (or if it is, it tends to be ignored as an inconvenient fact). The present work is compelling, and relates to other recent work from the same lab that argued that the time of movement onset can be chosen independently of the time for preparation to complete. The present work demonstrates that this is perhaps the normal state of affairs, with the RT reflecting a variety of history-based effects. This is in some ways inconvenient and annoying: it means that the RT should be used with caution as a metric of computation time. On the other hand, the present work opens up new avenues to studying the factors that determine when the 'movement trigger' is pulled, and how that process is or isn't influence by the progress of movement preparation. Both these contributions (the caution, and the new avenue) are important.*
*The manuscript is generally well written, but could be tightened and clarified in places. I found the 'Changes in motivation' section of the Discussion a bit rambling and confusing. I kept being confused regarding whether increased motivation was being proposed to be the same thing as a speeding of the computational process. I would have usually thought of these as different (e.g., I might really want to solve the problem faster, but there is a limit) but I can see others as thinking they are the same. This section read as a bit reactionary against a potential criticism I didn't really understand. The clearest part of the Discussion is definitely the conclusions, which lay out a clear set of statements regarding how the present findings should be interpreted.*

We greatly appreciate this thorough commentary regarding our manuscript. We have added several comments to our Discussion section to expand on some of the key points that were raised here. We have also attempted to revise the discussion to improve clarity and readability, particularly with regards to the “changes in motivation” section.